# The potential role of religiosity, psychological immunity, gender, and age group in predicting the psychological well-being of diabetic patients in Saudi Arabia within the Bayesian framework

**Nawal A. Al Eid[1], Boshra A. Arnout** [2,3]*, **Thabit A. Al-Qahtani[4], Slavica Pavlovic[5], Mohammed R. AlZahrani[6], Abdalla S. Abdelmotelab[7], Youssef S. Abdelmotelab[7]**

1 Department of Islamic Studies, College of Arts, Princess Nourah bint Abdulrahman University, Riyadh, Saudi Arabia, 2 Department of Psychology, King Khalid University, Abha, Saudi Arabia, 3 Department of Psychology, Zagazig University, Zagazig, Egypt, 4 Department of Learning and Structure, King Khalid University, Abha, Saudi Arabia, 5 Faculty of Science and Education, University of Mostar, Mostar, Bosnia and Herzegovina, 6 Department of Psychology, Umm Al-Qura University, Al-Abidiyah, Makkah, Saudi Arabia, 7 Faculty of Medicine, Badr University, Cairo, Egypt

* prof.arnout74@gmail.com, beahmad@kku.edu.sa

**Data Availability Statement:** All relevant data are within the manuscript.

## Abstract

This study aimed to investigate the differences in Religiosity (R), Mental Immunity (MI), and Psychological Well-Being (PWB) in patients with diabetes due to gender and age group variables, and to detect the best predictors of PWB in diabetic patients within the Bayesian framework. The study was conducted from May 2022 to February 2023 on a random sample of 186 Saudis diagnosed with diabetes. After obtaining participants' consent, they completed three R, MI, and PWB scales. Bayesian Independent Samples *t-test* was performed to identify differences, and Bayesian linear regression analysis was used to reveal the best prediction model of PWB. The results of the Bayesian independent samples *t-test* indicated strong evidence supporting the alternative hypothesis $H_1$, suggesting differences between male and female diabetic patients in R, MI, and PWB, with Bayesian factor values exceeding 10 ($8.338\times10^{+23}$, $1.762\times10^{+25}$, and $1.866\times10^{+24}$), and Cohen's δ of (-1.866, -1.934, -1.884). These results indicated that females with diabetes have higher means of R, MI, and PWB compared to males. However, the results also suggested evidence for the null hypothesis $H_0$ of no differences in R, MI, and PWB among diabetic patients due to age group, with Bayesian factor values (0.176, 0.181, and 0.187) less than 1.00 and small Cohen's δ of (-0.034, -0.050, -0.063). Bayesian linear regression analysis detected strong evidence that the model including MI is the best predictive model ($BF_{10}$ for mental immunity is 1.00 and for the other two models are 0.07 and $4.249\times10^{-16}$) for the PWB of diabetic patients, however, there is no evidence that the model including R or the interaction between R and MI is the best predictor of PWB for diabetic patients. These findings highlight the need for direct psychological care services for male diabetic patients and the urgent need to enhance IM in diabetic patients to

**Funding:** This work was supported by Princess Nourah bint Abdulrahman University Project number (PNURSP2023R380). The funders had no role in the study design, data collection, analysis, the decision to publish, or the preparation of the manuscript.

**Competing interests:** The authors declare that they have no conflict of interest.

**Abbreviations:** R, Religiosity; MI, Mental Immunity; PWB, Psychological Well-Being; $H_0$, Null hypothesis; $H_1$, Alternative hypothesis; R-S, Religiosity Scale; M-Q, Mental Immunity Questionnaire; PWB-S, PWB scale; EFA, Exploratory Factor Analysis; CFA, Confirmatory Factor Analysis; BF, Bayesian Factor.

improve their PWB. Furthermore, results recommended that healthcare providers in Saudi Arabia integrate MI interventions into diabetes care programs.

## 1. Introduction

Diabetes is a significant public health challenge in the 21st century. It is a metabolic disorder characterized by high blood sugar levels, resulting from a malfunction in carbohydrate metabolism, insufficient secretion of insulin by the pancreas, or the body's resistance to insulin, leading to elevated blood and urine sugar levels. It is a chronic disease that can have severe consequences if not properly managed. It is considered one of the most perilous chronic diseases due to its impact on health and the high mortality rate associated with it. It is crucial to adhere to medical advice and treatment plans to prevent complications and maintain good health [1,2].

Diabetes is a chronic disease, from which about 422 million people around the world suffer, while the number of cases and its prevalence will increase during the coming decades. Diabetes affects both the body and the soul. There is growing theoretical literature on the effect of diabetes on the mental health of individuals [1–4]. Deschênes et al. [2] reported that diabetes is considered one of the physical disorders where psychological factors contribute to worsening the condition, increasing neurological disorders such as constant fear, future anxiety, insomnia, memory impairment, increased agitation, loss of life enjoyment, thus affecting the psychological well-being of diabetics.

Many researchers and psychologists have long concentrated on the negative aspects of human personality, overlooking positive traits, and dealing with cognitive distortions, behavioral disorders, and psychological disturbances. A new trend has since emerged, shifting focus towards exploring sources of strength and human virtues like PWB, mental resilience, and religiosity [5].

The term religiosity refers to a multidimensional construction of beliefs, behaviors, and rituals that developed over time within societies, and which facilitate an understanding of an individual's relationship with God and his/her responsibility toward others [6]. Religiosity is the search for truth, and meaning in life to achieve inner peace and harmony, which allows the individual to achieve self-actualization and personal development [7–10].

In times of stress and disaster, religiosity provides individuals with happiness, security, reassurance, satisfaction, hope, optimism, and forgiveness for themselves and others. There is a growing body of literature highlighting that positive emotions enhance the immune system [5,6,8]. Additionally, there is a significant link between religiosity, pro-health behavior, and PWB [8].

The theoretical literature indicates a significant effect of religiosity on physical and mental health, but the mechanism of this effect is still understood well. One example of how religiosity can affect an individual's physical health and PWB is neuropsychological immunology (Koenig, 2000). Neuropsychological immunology is concerned with studying the effect of psychological and social factors on endocrine functions and body immunity, and one of its interests is studying how religiosity affects immune function in stressful and catastrophic situations [8].

Moreover, the feelings of sadness, fear, and psychological distress resulting from the individual's trauma due to his/her awareness of having diabetes affect the immune system negatively, thus deteriorating the physical health and PWB of diabetic patients [11,12].

Similarly, the previous studies found that religiosity was considered as a coping stress mechanism, and that frequency of participation in religious activities was a significant

predictor of subsequent cortisol levels [13]. Also, religiosity is a significant predictor of an attitude to alcohol abuse [14]. Religiosity is related to mental and physical health [15]. Recently, many studies have found religiosity moderates the effects of COVID-19 and maintains spiritual, mental, and physical resilience through the COVID-19 outbreak [16].

Moreover, theoretical background refers to the mental immune system as an integrated system of cognitive, motivational, and behavioral dimensions, that enhance resistance against stress, promote healthy growth, and act as stress-resistance resources or mental antibodies [17].

According to Oláh [18] there are three sub-systems: approach-belief, the control-creation-executive, and the self-regulation sub-system, which constitute a multidimensional structure that provides immunity against stress and trauma through the continuous exploration of the transient environment and the absorption of unique experiences. Therefore, the mental immune system creates a balance between personality and environmental factors to increase resilience.

Regarding the PWB, there are many perspectives; in 1998, Ryff presented a model of PWB that includes six dimensions: autonomy, self-acceptance, life purpose, social relationships, environmental empowerment, and personal maturity. Autonomy refers to the feeling of an individual's independence; self-acceptance means that the individual accepts himself/herself with his positives and negatives. The purpose of life refers to a person's beliefs about his/her life in terms of its purpose and meaning. Social relations refer to those positive, warm, purposeful relationships, while environmental empowerment means the ability to manage the environment. In this study, we developed PWB for diabetics in light of Ryff's model of PWB. However, Bozek et al. [19], AL-Zahrani [20], AL-Najjar [21], Arnout & Almoied [22], Hadi [23], and Ryff & Singer [24] argued that PWB consists of personal maturity, self-efficacy, openness, positive self-evaluation, the meaning of life, positive relationships, manage the life, control over the environment, self-determination, achieve goals, contentment, joy, optimism, self-realization, self-confidence, and stress management.

In addition, Dodge [25] saw the PWB as a point of balance between personal, social, and physical resources that enable us to face life difficulties and achieve life harmonization by employing those resources. Thus, Dodge argued that if the individual possesses large resources, he/she succeeds in coping with the stress of life, while if the person's resources shortcut, both his/her ability to cope and his PWB will decrease also.

Concerning the PWB of diabetes, the study by Al-Ghdani & Sskamanya [26] revealed that the PWB of diabetic patients in Oman is moderately acceptable across all dimensions. It was observed that the PWB of diabetic patients post-diagnosis was perceived as poor, with common complications including visual impairment, sexual dysfunction, foot disease, kidney disorders, eating disorders, and heart and artery issues. Differences in PWB levels among diabetic patients in Oman were linked to gender differences, while no disparities were associated with the type of treatment and duration of diabetes. In the same context, Mark et al. [27] showed a link between diabetes anxiety and PWB. Additionally, Reitimeir et al. [28] found that there were differences in the level of PWB among diabetic patients attributed to gender in favor of males and the duration of diabetes in favor of shorter duration. Also, Al-Anzi's [29] study results have shown that chronic diseases like diabetes have a negative impact on mental well-being.

Bayesian framework is a statistical method that uses Bayesian theory to analyze data and estimate parameters. It is a predictive statistical method to revise the probability of a hypothesis when obtaining new evidence or information, theoretical literature states that it provides many benefits, such as considering prior knowledge to create a clearer hypothesis test, the ability to monitor evidence for the null or alternative hypothesis as data accumulates, and distinguishing between the presence or absence of evidence [30–32]. For example, Castillo et al. [33]

indicated that Bayesian linear regression deduces model parameters, as it offers superior outcomes compared to the maximum likelihood estimation method. This advantage stems from its ability to facilitate probabilistic interpretations of model coefficients and furnish a posterior distribution for the model parameters, grounded on the data utilized for inferring about the parameters. Also, Consonni et al. [34] reported that Bayesian linear regression is characterized by the integration of prior knowledge regarding the unknown parameters, Bayesian estimation thus surpasses the maximum likelihood estimation method, particularly in scenarios involving small sample sizes.

From the viewpoint that scientific specialization's capacity depends on accumulating knowledge through data analysis, Bayesian statistics plays a crucial role in establishing a cumulative scientific framework that merges prior with posterior data. The Bayesian framework presents a unique perspective on hypothesis testing, enabling researchers to integrate foundational data into their analyses instead of frequently statistics on the same null hypothesis [35]. Therefore, Bayesian methods excel over frequentist methods in statistics [31]. However, a challenge of Bayesian statistics is the selection of the prior distribution for the data, as it involves translating subjective beliefs into mathematically formulated prior beliefs, often at a significant computational cost, the availability of computational tools in common software packages such as Amos, Bayesian methods in Mplus, various packages in the statistical computing environment R, and statistical computing JASP has made conducting Bayesian analysis more cost-effective, user-friendly, and precise [31,35].

## 2. The current study

Previous studies have recommended further examination of religiosity and healthy behavior's role in PWB [19]. To our knowledge, there are no studies investigating the role of religiosity and mental immunity in predicting the PWB of diabetes patients within the Bayesian Framework. Thus, the main objective of this study is to test the evidence of the hypothesis about the differences in religiosity, mental immunity, and PWB due to gender and age group, and also, the evidence of the hypothesis about the best model to predict PWB among diabetes patients. Therefore, the current study sought to test the following:

### Hypothesis 1

$H_1$. There were significant statistical differences between males and females in R, MI, and PWB of diabetic patients in favor of females.

### Hypothesis 2

$H_1$. There were significant statistical differences in R, MI, and PWB of diabetic patients due to age group, in favor of diabetes patients younger than 50 years old group.

### Hypothesis 3

$H_1$. Religiosity and mental immunity predict the PWB of diabetic patients in the Kingdom of Saudi Arabia.

## 3. Methodology

### 3.1. Method and participants

The current study applied the cross-sectional descriptive design to test the predictability of PWB in patients with diabetes from religiosity and mental immunity after the Research Ethics Committee at Princess Nourah bint Abdulrahman University approved this study, number

**Table 1. Participants' demographic characteristics.**

| Sociodemographic Variable | N% |
|---|---|
| Participants | 186 |
| **Gender** | |
| Male | 84 (45.16%) |
| Female | 102 (54.84%) |
| **Region** | |
| Riyadh | 86 (46.24%) |
| Makkah Al-Mukarramah | 34 (18.28%) |
| Southern Province | 66 (35.48%) |
| **Nationality** | |
| Saudi | 144 (77.42%) |
| Non-Saudi | 42 (22.58%) |

(23–0493). All methods were carried out by relevant guidelines and regulations, and written informed consent was obtained from all subjects to participate in the current study. The study sample consisted of 186 patients who were diagnosed with diabetes and visited the outpatient clinics of hospitals in Riyadh, Makkah-Al-Mukarramah, and Southern Provin in Saudi Arabia, through the period from 19 May 2022 to 8 February 2023. Those participants were selected randomly, 84 (45.16%) of them were men and 102 (54.84) were women, and their ages ranged between 42 to 57 years old (48.01±4.11), 144 (77.42%) were Saudi, and 42 (22.58%) non-Saudi, who were living in Riyadh (n = 86, 46.24%), Makkah Al-Mukarramah (n = 34, 18.28%), and Southern Provin (n = 66, 35.48%) (Table 1).

## 3.2. Measurements

The score of religiosity was assessed using the Religiosity Scale (R-S). R-S is an 8-item self-report tool assessing religiosity on a 5-point scale. In addition, the Mental Immunity Questionnaire (MI-Q) consisted of 8 items, to examine the score of mental immunity by using a 5-point scale. Also, the PWB scale (PWB-S) consisted of 9 items and was used to measure the scores of the participants' PWB, by using a 5-point scale. All measurements, R-S, MI-S, and PWB-S were prepared by the researchers in this study, after reviewing the theoretical literature, the initial version of each scale was prepared and presented to evaluators specialized in psychology and psychometrics, and amendments were made to the initial version of the scale in light of their comments.

To verify the validity of the R-S, MI-S, and PWB-S, researchers used Exploratory Factor Analysis (EFA) by the Principal Components method for each scale item, rotating them orthogonally using Promax to extract the structural factorial of these scales, after confirming the adequacy of the analysis sample through the Kaiser-Meyer-Olkin Measure (0.940, 908, and 0.952) for R-S, MI-Q, and PWB-S respectively. The following Table 2 illustrate the results of the EFA for the three scales after rotation.

The EFA of the Religiosity Scale (R-S) revealed one factor that all scale items saturated, with saturation values ranging between (0.838–0.936), with eigenvalue (6.479), and total variance (78.30%), this factor can be named "General Religiosity" Factor. Similarly, the EFA of the Mental Immunity Questionnaire (MI-Q) showed all items saturating on one factor with an eigenvalue of (6.549), saturation values ranging between (0.856–0.928), and a total variance of (79.30%), which can be named the "General mental immunity" factor. Furthermore, the EFA of the Psychological Well-being scale (PWB-S) revealed one general factor "Psychological

**Table 2. Factor loading for R-S, MI-Q, and, PWB-S.**

| R-S Items | Factor 1 | MI-S Items | Factor 1 | PWB-S Items | Factor 1 |
|---|---|---|---|---|---|
| R6 | 0.936 | I4 | 0.928 | W7 | 0.937 |
| R4 | 0.934 | I7 | 0.925 | W1 | 0.929 |
| R2 | 0.903 | I5 | 0.922 | W3 | 0.913 |
| R3 | 0.895 | I3 | 0.893 | W4 | 0.913 |
| R8 | 0.869 | I2 | 0.880 | W2 | 0.907 |
| R5 | 0.854 | I6 | 0.860 | W6 | 0.906 |
| R7 | 0.847 | I1 | 0.856 | W5 | 0.899 |
| R1 | 0.838 | I8 | 0.856 | W9 | 0.884 |
| Factor Characteristics | | | | W8 | 0.883 |
| Eigenvalues | 6.479 | 6.549 | | 7.596 | |
| Total Variance % | 0.783 | 0.793 | | 0.825 | |

The applied rotation method is Promax.

Well-being" with an eigenvalue of (7.596), saturation values ranging between (0.883–0.937), and total variance of (82.50%). The following Fig 1 illustrates the results after rotation using the Promax method and extracting the saturation results of R-S, MI-Q, and PWB-S items using the JASP 0.18.3.0 Program.

The structural factorial of the components of R-S, MIQ, and PWB-S is shown in Fig 1, with the dark green lines indicating strong relations and saturations between each scale's items and the general factor.

Also, Confirmatory Factor Analysis (CFA) was utilized to verify the standard model of R-S, MIQ, and PWB-S. The goodness-of-fit indices fell within the good indices, and the following Table 3 shows the results of CFA for each scale.

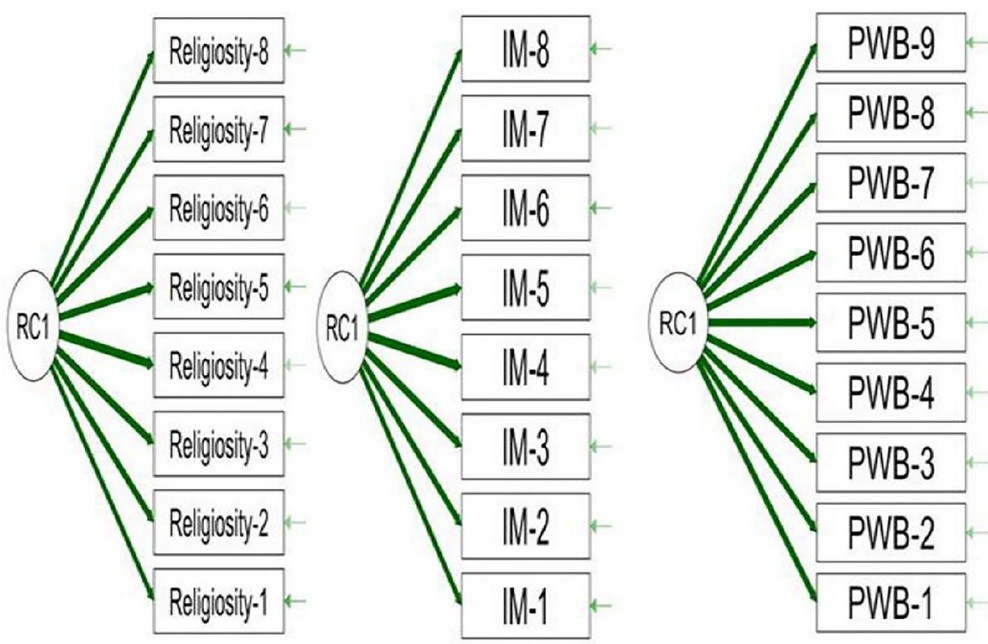

**Fig 1. EFA results after rotation for items of R-S, MIQ, and PWB-S.**

**Table 3. Fit indices for each of R-S, MIQ, and PWB-S.**

| Index | Value | | |
|---|---|---|---|
| | **R-S** | **MI-Q** | **PWB-S** |
| **Comparative Fit Index (CFI)** | 0.939 | 0.921 | 0.966 |
| **Tucker-Lewis Index (TLI)** | 0.914 | 0.890 | 0.955 |
| **Bentler-Bonett Non-normed Fit Index (NNFI)** | 0.914 | 0.890 | 0.955 |
| **Bentler-Bonett Normed Fit Index (NFI)** | 0.929 | 0.912 | 0.955 |
| **Parsimony Normed Fit Index (PNFI)** | 0.664 | 0.651 | 0.716 |
| **Bollen's Relative Fit Index (RFI)** | 0.901 | 0.877 | 0.940 |
| **Bollen's Incremental Fit Index (IFI)** | 0.939 | 0.922 | 0.966 |
| **Relative Noncentrality Index (RNI)** | 0.939 | 0.921 | 0.966 |

CFA results shown in Table 3 indicate that the fit indices for the standard model of each scale of R-S, MIQ, and PWB-S fell within acceptable indices. Additionally, the path coefficients for all items composing the latent factor were all statistically significant (0.001) (Fig 2), thus, results confirming the one-factorial standard model for R-S, MIQ, and PWB-S.

Path coefficients shown in Fig 2, indicated that the path coefficient between the items and their scale are statistically significant. The path coefficients for the R-S ranged between (0.0.90–1.14), and for the IM-Q ranged between (0.94–1.21), while for the PWB-S ranged between (0.97–1.17). These results indicated the theoretical model's validation and its alignment with the data collected from the sample participants, implying the congruence of the

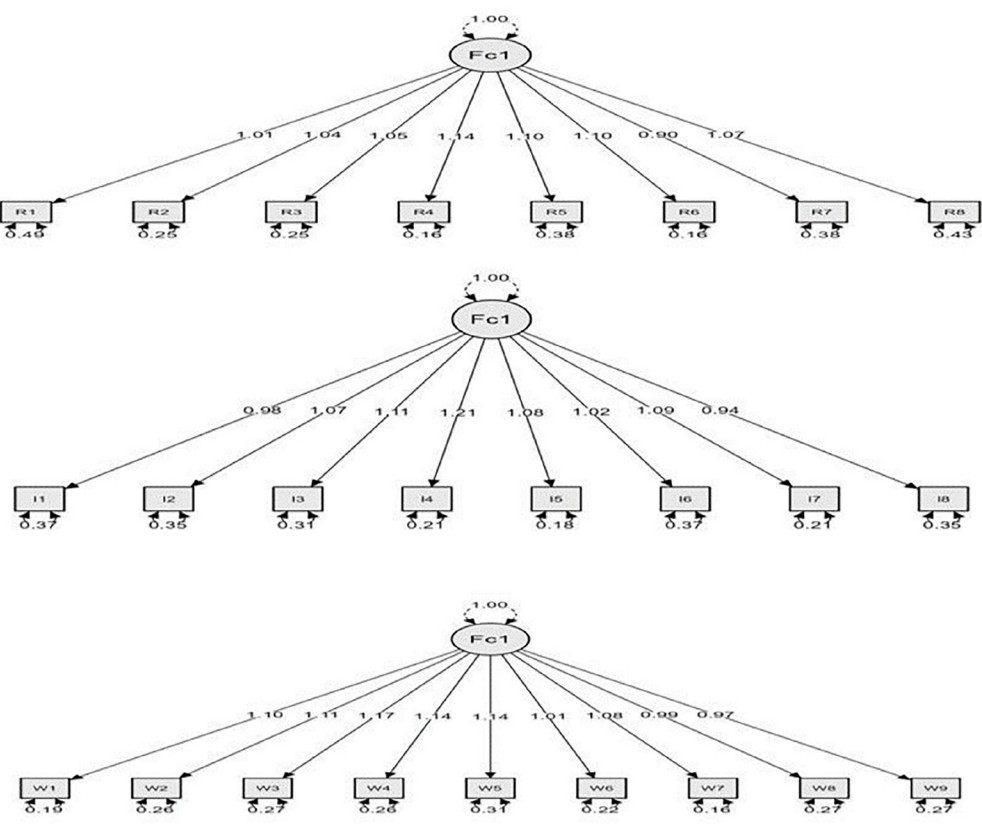

**Fig 2. CFA results for R-S, MIQ, and PWB-S using JASP software.**

**Table 4. Correlations between the item R-S, MIQ, and PWB-S and reliability coefficients.**

| Scale | Religiosity | | Mental Immunity | | PWB | |
|---|---|---|---|---|---|---|
| | Item | Correlation | Item | Correlation | Item | Correlation |
| | 1 | .756** | 1 | .878** | 1 | .934** |
| | 2 | .768** | 2 | .899** | 2 | .919** |
| | 3 | .782** | 3 | .909** | 3 | .925** |
| | 4 | .808** | 4 | .934** | 4 | .924** |
| | 5 | .769** | 5 | .928** | 5 | .913** |
| | 6 | .824** | 6 | .881** | 6 | .916** |
| | 7 | .766** | 7 | .929** | 7 | .940** |
| | 8 | .836** | 8 | .877** | 8 | .897** |
| | | | | | 9 | .898** |
| **Cronbach's α** | .973 | | .968 | | .976 | |
| **Spearman-Brown Coefficient** | .979 | | .949 | | .966 | |
| **Guttman Split-Half Coefficient** | .979 | | .948 | | .939 | |

**Correlation is significant at 0.01 level.

CFA results with the EFA results and the alignment of the proposed one-factorial standard model with the empirical data collected.

In addition, we tested the reliability of the R-S, MI-Q, and PWB-S, results revealed that the R-S, MI-Q, and PWB-S had acceptable internal consistency. Whenever testing internal consistency and Cronbach's α for all items loadings was greater than 0.70, The value of Cronbach's α reliability coefficients was for R-S (0.973), MI-Q (0.968), and PWB-S (0.976) (Table 4).

## 4. Data analysis

Given the nature of psychological studies, the Bayesian framework is most suitable to test this study's hypotheses because it incorporates prior data into the statistical model. Bayesian inference offers direct probabilistic information about parameters, which is more beneficial for scientific research compared to the confidence levels provided by frequentist statistics. JASP 0.18.3.0 program [36] was used for data statistical analysis. Researchers utilized the Bayesian Independent Samples *t-test* to test the hypotheses of the mean differences in R-S, MI-Q, and PWB-S due to gender and age group variables. Also, the Bayesian linear regression is employed for predicting PWB of diabetic patients from religiosity (R) and mental immunity (MI). Based on the Jeffreys criteria for the Bayesian Factor, If BF between (1 and 3) is weak, and between (3 and 10) is median, while between if (greater than 10) is strong [37,38]. Researchers utilized an informative prior JASP setting for Cauchy r (0.707) in favor of the alternative hypothesis compared to the null hypothesis.

## 5. Results

### 5.1. Results of the hypothesis 1

The Bayesian Independent-Samples *t-test* was used to test the probability of the data by comparing the alternative hypothesis $H_1$ (there were significant statistical differences between males and females in R, MI, and PWB of diabetic patients in favor of females) compared to the null hypothesis $H_0$ (there were no differences between males and females in R, MI, and PWB of diabetic patients), as it provides a measure of evidence and continuously monitors its strength by collecting data. Results are shown in Tables 5 and 6.

**Table 5. Bayesian independent samples *t-test*.**

| Measures | $BF_{10}$ | error % |
|---|---|---|
| Religiosity | $8.338 \times 10^{+23}$ | $7.930 \times 10^{-30}$ |
| Mental Immunity | $1.762 \times 10^{+25}$ | $4.624 \times 10^{-31}$ |
| Psychological Well-being | $1.866 \times 10^{+24}$ | $3.788 \times 10^{-30}$ |

The results shown in Tables 5 and 6, indicated the Bayes factor ($BF_{10}$) of ($8.338 \times 10^{+23}$, $1.762 \times 10^{+25}$, and $1.866 \times 10^{+24}$) for R, MI, and PWB respectively suggests that the $BF_{10}$ were extremely large (more than 10) about ($8.338 \times 10^{+23}$, $1.762 \times 10^{+25}$, and $1.866 \times 10^{+24}$) times and so, in this case, we can say our $BF_{10}$ in favor of the alternative hypothesis ($H_1$) that there were mean differences between males and females in R, MI, and PWB, respectively, in compare to the null hypothesis ($H_0$) of no mean defenses between males and females in R, MI, and PWB

Also, Fig 3–5 concluded that the prior is very flat and wide, shifted posteriorly to the left (negative), providing greater density than before for the three variables: R, MI, and PWB of diabetic patients. Therefore, the data strongly suggest that female diabetic patients have higher means in R, MI, and PWB compared to males. This is supported by the density on the X-axis and the effect size for the *t-test* (Cohen's δ) in this case were large effects (-1.866, -1.934, -1.884) based on Cohen (1988), and were reaching their peak with a reliable 95% confidence interval at (-2.217, -1.516; -2.289, -1.581; -2.236, -1.533) for R, MI, and PWB respectively.

In addition, the robustness test of the $BF_{10}$ plots shown in Fig 6–8 indicates that the maximum $BF_{10}$ is attained when setting the prior width r to (1.5). The $BF_{10}$ robustness plot indicates $BF_{10}$ for the user prior (8.338e+23, 1.762e+25, 1.866e+24), wide prior (1.046e+24, 2.225e+25, 2.345e+24), and ultrawide prior (1.207e+24, 2.595e+25, 2.715e+24) for R, MI, and PWB respectively. These findings revealed the evidence for $H_1$ is relatively stable across a wide range of prior distributions, suggesting that the analysis is robust. However, the evidence in favor of $H_0$ is not particularly strong. Furthermore, another evidence, Cohen's δ of (-1.866, -1.934, -1.884) with a reliable 95% confidence interval indicates significant effects based on Cohen's rule (1988), where female diabetic patients exhibit a higher level of R, MI, and PWB compared to males.

## 5.2. Results of the hypothesis 2

The Bayesian Independent-Samples *t-test* was used to measure the probability of the data by comparing $H_1$ with $H_0$, with an informative prior JASP setting for Cauchy r (0.707) in favor of $H_1$ (there were significant statistical differences in R, MI, and PWB of diabetic patients due to age group, in favor of diabetes patients younger than 50 years old) compared to $H_0$ (there were

**Table 6. Descriptives.**

| Measures | Group | N | Mean | SD | SE | Coefficient of variation | 95% Credible Interval Lower | Upper |
|---|---|---|---|---|---|---|---|---|
| Religiosity | Males | 101 | 18.990 | 6.888 | 0.685 | 0.363 | 17.630 | 20.350 |
| | Females | 84 | 30.810 | 5.365 | 0.585 | 0.174 | 29.645 | 31.974 |
| Mental Immunity | Males | 101 | 18.881 | 6.795 | 0.676 | 0.360 | 17.540 | 20.223 |
| | Females | 84 | 31.048 | 5.411 | 0.590 | 0.174 | 29.873 | 32.222 |
| Psychological Well-being | Males | 101 | 20.871 | 7.649 | 0.761 | 0.367 | 19.361 | 22.381 |
| | Females | 84 | 34.488 | 6.439 | 0.703 | 0.187 | 33.091 | 35.885 |

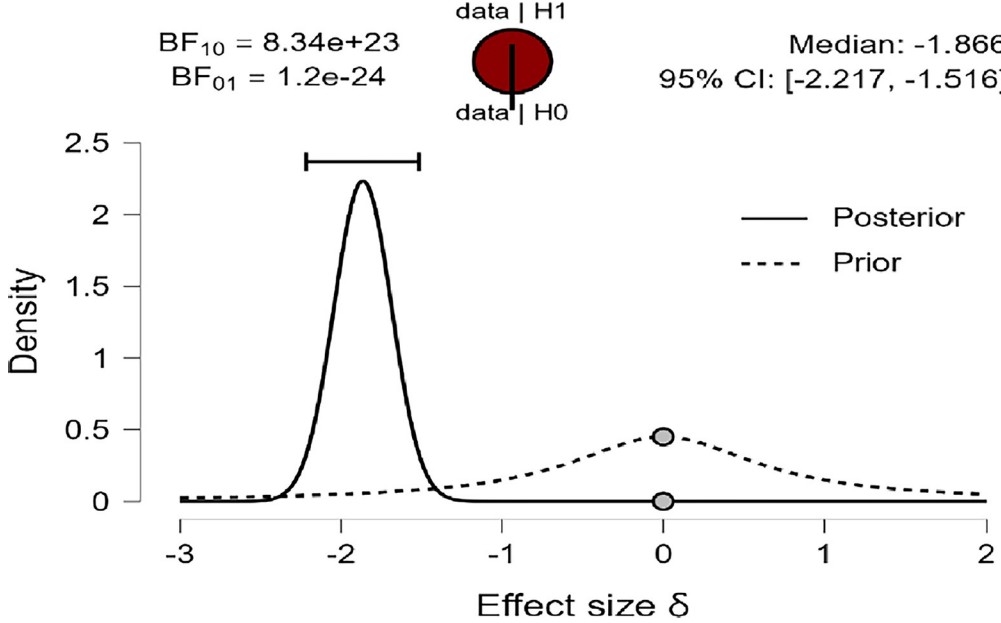

**Fig 3. Prior and posterior religiosity.**

no differences in R, MI, and, PWB of diabetic patients due to age groups variable), results shown in Tables 7 and 8.

The results of the Bayesian Independent Sample *t-test* are shown in Tables 7 and 8, indicating the Bayes factor ($BF_{10}$) of (0.176, 0.181, and 0.187) for R, MI, and PWB respectively suggesting that the Bayes Factor were very low (less than 1.00), and so, in this case, we just can say our $BF_{10}$ in favor of $H_0$ that there were no mean differences in R, MI, and PWB due to age

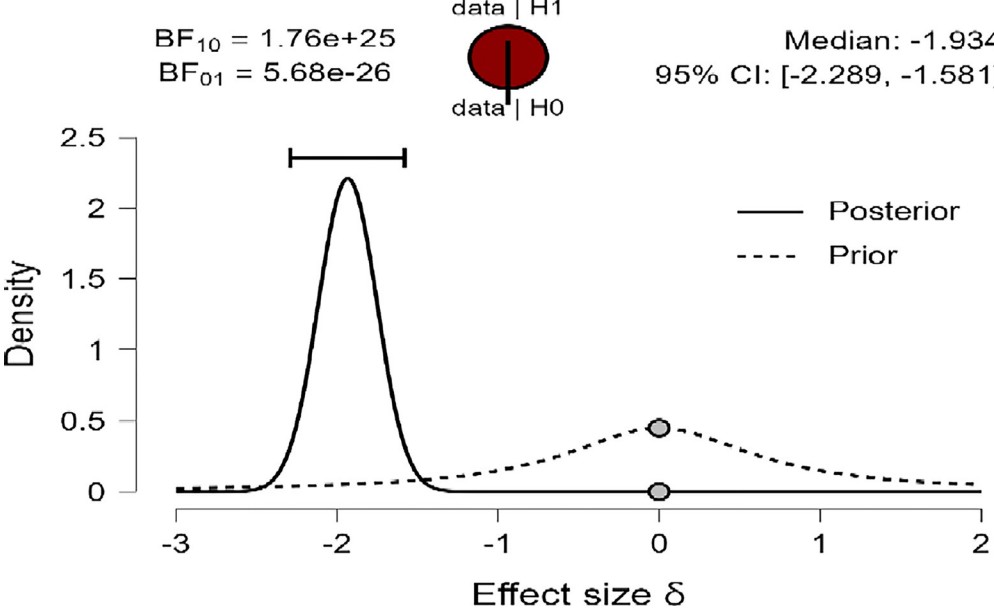

**Fig 4. Prior and posterior for mental immunity.**

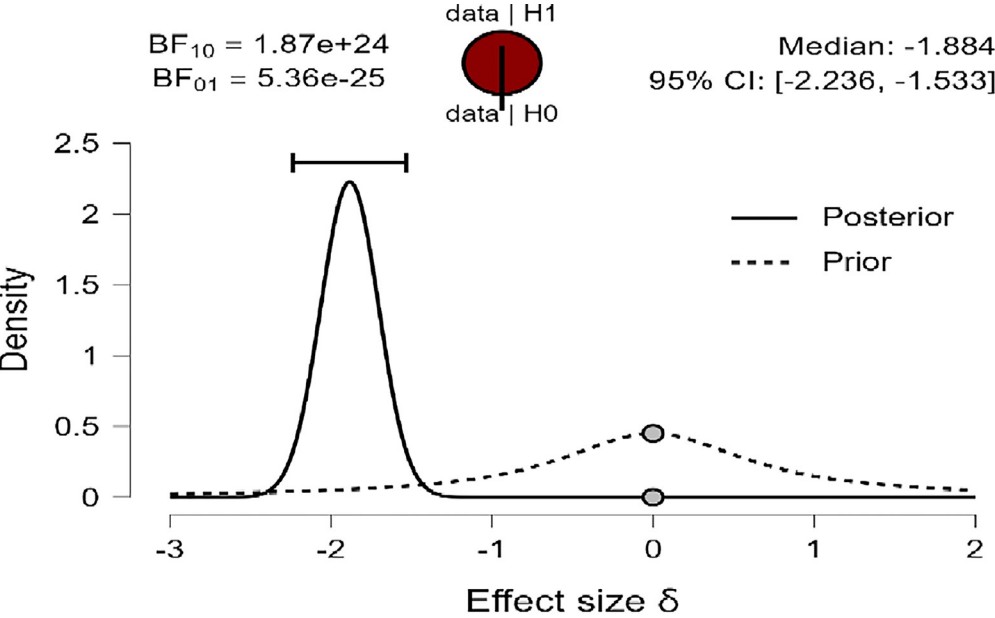

**Fig 5. Prior and posterior for PWB.**

group variable, in compare to $H_1$ of there were mean differences in R, MI, and PWB due to age group in favor of diabetes patients younger than 50 years old group.

Also, Figs 9–11 concluded that the prior is very flat and wide, and not shifted posteriorly for the three variables: R, MI, and PWB of diabetic patients. Therefore, the data strongly suggest $H_0$ that there are no differences in R, MI, and PWB due to age group variables. This is

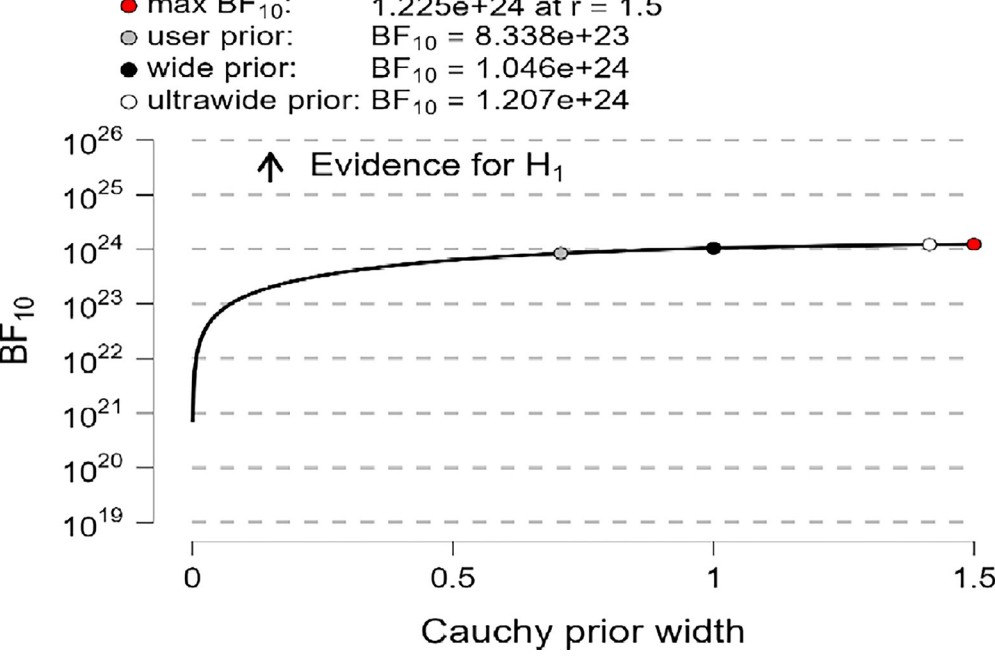

**Fig 6. Bayes factor robustness check for religiosity.**

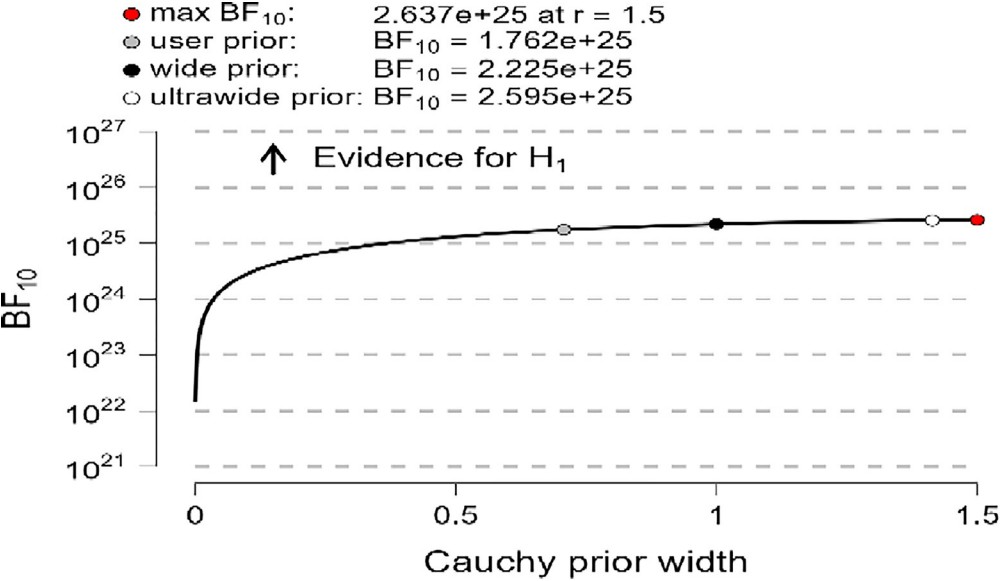

**Fig 7. Bayes factor robustness check for mental immunity.**

supported by the effect size for the *t-test* (Cohen's δ) in this case were small effects (-0.034, -0.050, -0.063) based on Cohen (1988), and were reaching their peak with a reliable 95% confidence interval at (-0.334, 0.265; -0.350, 0.52481; -0.364, 0.235) for R, MI, and PWB respectively.

In addition, the robustness test of $BF_{10}$ plots shown in Figs 12–14 indicates that the maximum $BF_{10}$ is attained when setting the prior width r to (5e-04). The $BF_{10}$ robustness plot

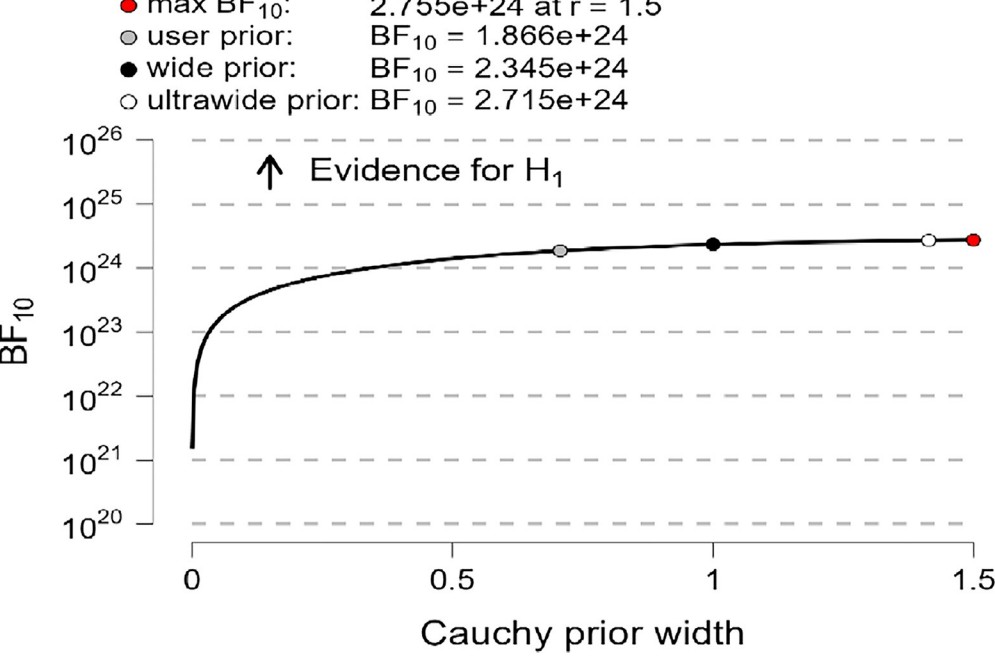

**Fig 8. Bayes factor robustness check for PWB.**

**Table 7. Bayesian independent samples *t-test*.**

| Measures | BF$_{10}$ | error % |
|---|---|---|
| Religiosity | 0.176 | $7.930 \times 10^{-30}$ |
| Mental Immunity | 0.181 | $4.624 \times 10^{-31}$ |
| Psychological Well-being | 0.187 | $3.788 \times 10^{-30}$ |

indicates BF$_{10}$ for the user prior (0.1761, 01815, and 0,1874), wide prior (0.1274, 0.1314, and 0.1358), and ultrawide prior (0.09117, 0.09411, and 0.09737) for R, MI, and PWB respectively. These findings revealed the evidence for H$_0$ is relatively stable across a wide range of prior distributions, suggesting that the analysis is robust. However, the evidence in favor of H$_1$ is not particularly strong. Also, another evidence, Cohen's δ of (-0.034, -0.050, -0.063) with a reliable 95% confidence interval indicates no significant effects based on Cohen's rule (1988), where no differences between younger than 50 years old diabetic patients group, compared to the diabetes patients group whom their ages ranged between 50 years old and over in R, MI, and PWB.

## 5.3. Results of the hypothesis 3

The Bayesian Linear Regression model was employed to identify the best predictors of PWB in diabetic patients, to measure the probability of the data by comparing H$_1$ with H$_0$, with an informative prior JASP setting for Cauchy r (0.707) in favor of H$_1$ (Religiosity and mental immunity predict the PWB of diabetic patients in the Kingdom of Saudi Arabia), compared to H$_0$ (Religiosity and mental immunity do not predict the PWB of diabetic patients in the Kingdom of Saudi Arabia. The Bayesian Linear Regression results are shown in Tables 9 and 10.

The results in Fig 15 about the Bayesian Correlation Matrix Plot of R, MI, and PWB, indicated that there is a linear relationship between R, MI, and PWB in diabetic patients. Also, the results show in Fig 16 the residuals against predictions for the PWB of diabetic patients dataset, and the red line representing the average value at each point. These results confirm the validation of the linear regression assumption.

Table 9 comparing four models for predicting the PWB of diabetic patients ranks these models from highest to lowest in terms of prediction accuracy. The results indicated that the data from the study sample most closely aligns with the model that MI is a predictor of PWB in diabetic patients, assigning it a posterior probability of (0.878). The next model (R + MI) has a posterior probability of (0.122), while the last two models have a combined posterior probability of $(3.729 \times 10^{-16}) + (9.680 \times 10^{-75})$, making them less effective for predicting PWB in diabetic patients. Table 8 further clarifies, through Bayesian factor values, that after post-data monitoring, the probabilities for the model MI increased from (35.913) to $(2.681 \times 10^{+15})$. This

**Table 8. Descriptives.**

| Measures | Group | N | Mean | SD | SE | Coefficient of variation | 95% Credible Interval Lower | 95% Credible Interval Upper |
|---|---|---|---|---|---|---|---|---|
| Religiosity | Younger than 50 years old | 128 | 24.258 | 8.610 | 0.761 | 0.355 | 22.752 | 25.764 |
| | 50 years old and over | 57 | 24.579 | 8.579 | 1.136 | 0.349 | 22.303 | 26.855 |
| Mental Immunity | Younger than 50 years old | 128 | 24.258 | 8.680 | 0.767 | 0.358 | 22.740 | 25.776 |
| | 50 years old and over | 57 | 24.737 | 8.719 | 1.155 | 0.352 | 22.423 | 27.050 |
| Psychological Well-being | Younger than 50 years old | 128 | 26.844 | 9.850 | 0.871 | 0.367 | 25.121 | 28.567 |
| | 50 years old and over | 57 | 27.526 | 9.869 | 1.307 | 0.359 | 24.908 | 30.145 |

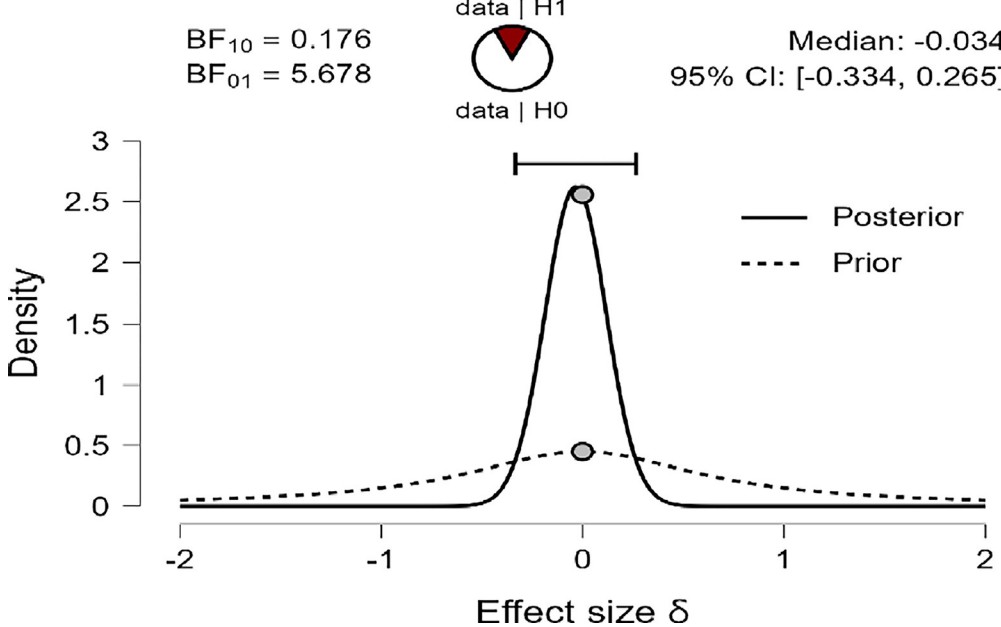

**Fig 9. Prior and posterior religiosity.**

underscores the superior predictive capability of the MI model, establishing it as a more effective predictor of PWB scores in diabetic patients ($BF_{10}$ for mental immunity is 1.00, and for the other two models is 0.07 and $4.249 \times 10^{-16}$).

As well as, the results presented in Table 10 and Fig 17 demonstrate that the marginal posterior probability distribution for the religiosity coefficient shows a pronounced spike at a value

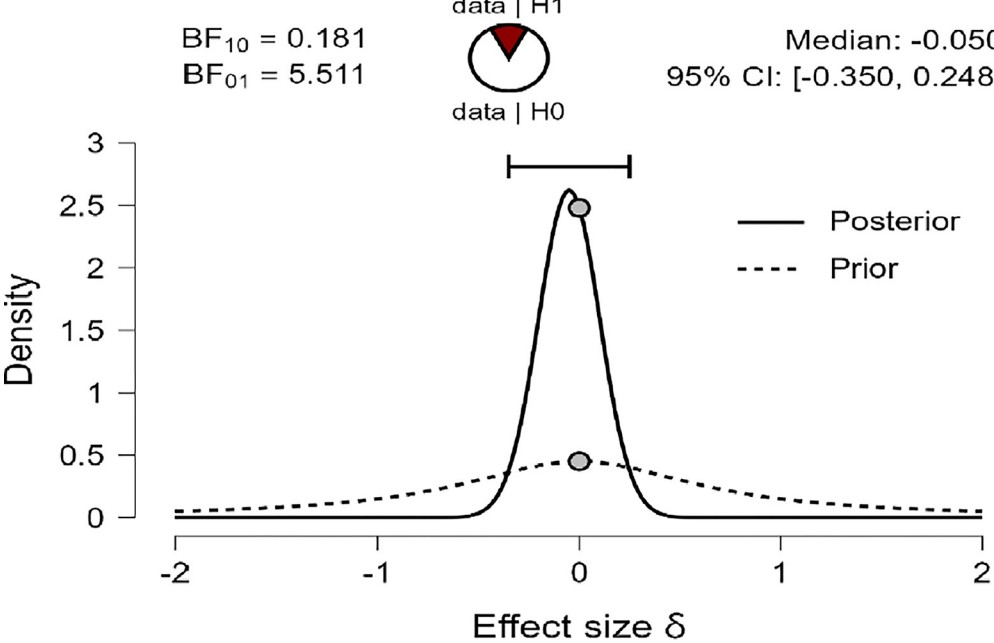

**Fig 10. Prior and posterior for mental immunity.**

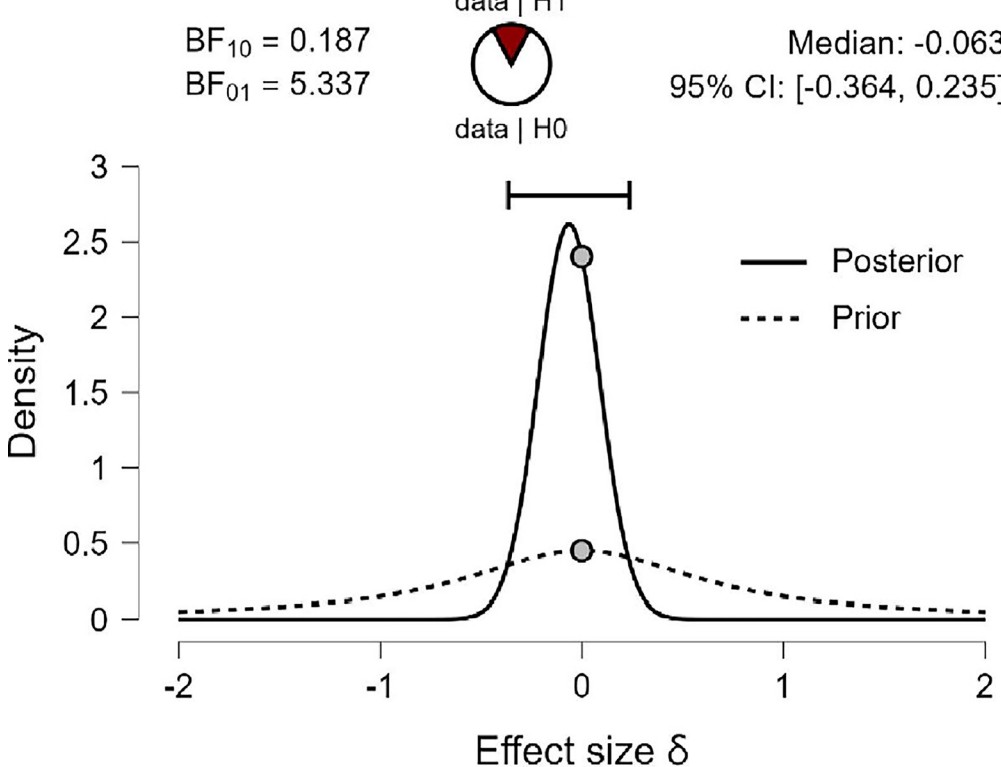

**Fig 11. Prior and posterior for PWB.**

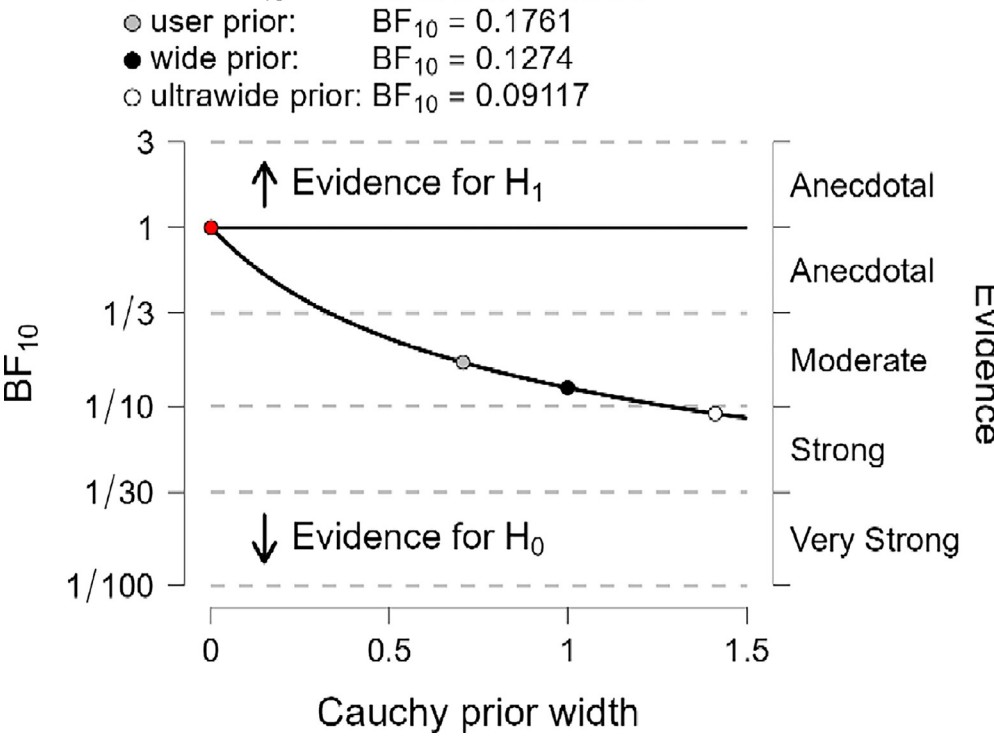

**Fig 12. Bayes factor robustness check for religiosity.**

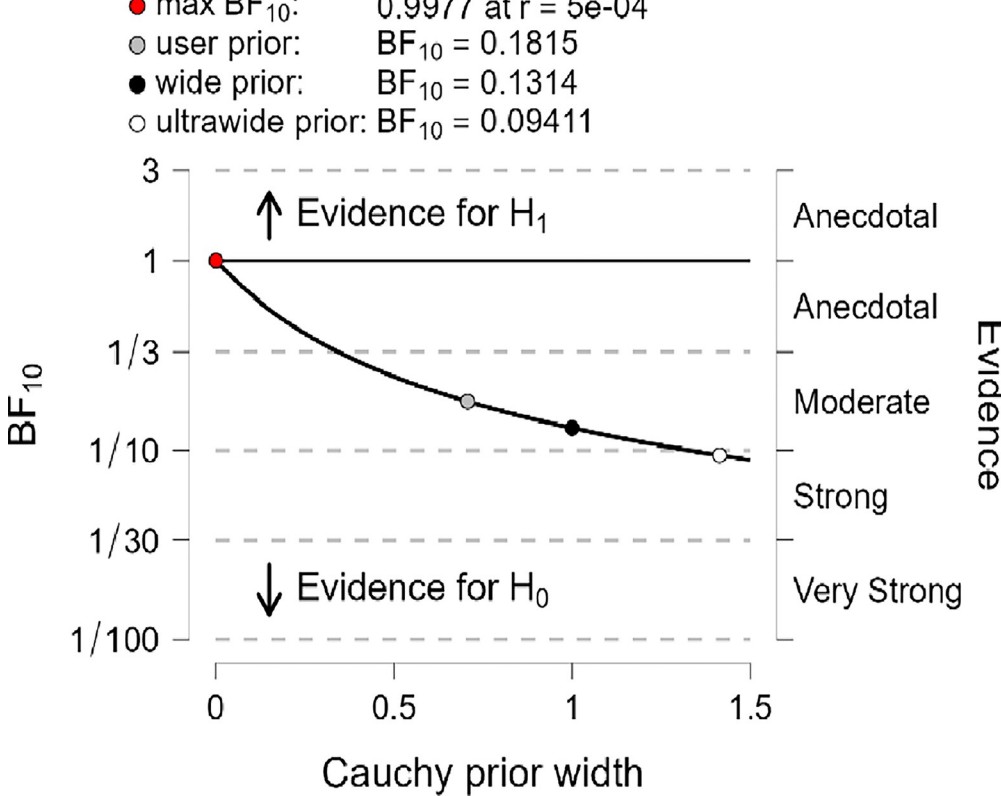

**Fig 13. Bayes factor robustness check for mental immunity.**

of zero for R, indicating a high probability of (0.878) for the exclusion of R as a predictive variable in the model assessing the PWB of diabetic patients. When R is considered in the model (with a probability of inclusion at (0.878)), there is a 95% probability that its impact fluctuates between -0.264e-03 point and +1.75e-01 point. Consequently, we do not witness a consistent influence of religiosity, suggesting there is substantial evidence to support the absence of any significant effect. While we included MI in the model, there is a 95% probability that its impact fluctuates between + 0.921 point and +1.128 point (Figs 18 and 19).

## 6. Discussion

This study sought to test the evidence of the hypothesis about the differences in religiosity, mental immunity, and PWB due to gender and age group, and also, the evidence of the hypothesis about the best model to predict PWB among diabetes patients.

The results of the Bayesian independent samples t-test found strong evidence supporting differences between male and female diabetic patients in R, MI, and PWB, with $BF_{10}$ values exceeding 10 ($8.338 \times 10^{+23}$, $1.762 \times 10^{+25}$, and $1.866 \times 10^{+24}$), and Cohen's δ of (-1.866, -1.934, -1.884). These results indicated that females with diabetes have higher means of R, MI, and PWB compared to males, in favor of females. This result is consistent with the findings of Miller & Hoffmann [39] that females are more religious than males, and that preference for risk is a strong predictor of R for both males and females. The possible reason to explain these differences in R due to gender may be the socialization of females in Arab societies, which urges them to adhere to behavior. The woman turns to God a lot of the time in distress and

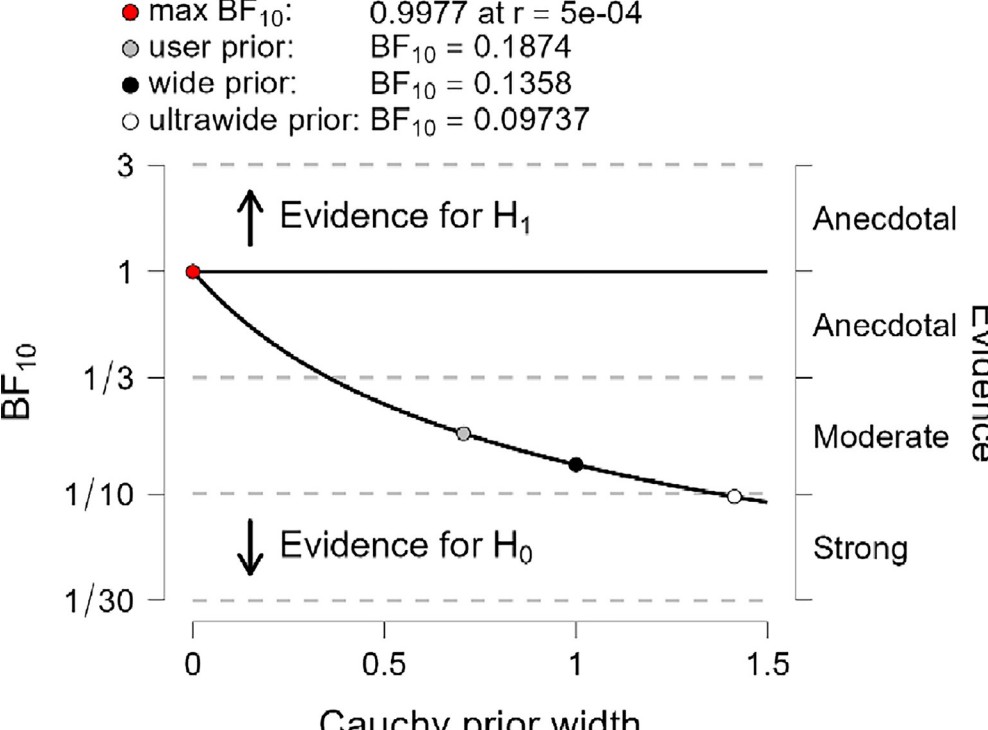

**Fig 14. Bayes factor robustness check for PWB.**

prays and blesses God. The nature of Arab women is religious and keen on a good relationship with God [40,41].

The results also revealed that there were statistically significant differences in MI, in favor of females. This is consistent with the results of studies [42,43] that indicated differences between males and females in MI in favor of males. However, it differs from the results of [44] Al-Saqa & Nadar's study, which indicated no differences in MI due to gender.

Additionally, the results found statistically significant differences in PWB in favor of females. This result is consistent with previous studies [40,44–46] that found statistically significant differences in PWB in favor of females. However, it is not consistent with the results of Reitimeir [27] who found there were statistically significant differences in PWB among

**Table 9. Model comparison—psychological well-being.**

| Models | P(M) | P(M\|data) | $BF_M$ | $BF_{01}$ | $R^2$ |
|---|---|---|---|---|---|
| Mental Immunity | 0.167 | 0.878 | 35.913 | 1.000 | 0.852 |
| Mental Immunity + Religiosity | 0.333 | 0.122 | 0.278 | 0.070 | 0.853 |
| Religiosity | 0.167 | $3.729\times10^{-16}$ | $1.865\times10^{-15}$ | $4.249\times10^{-16}$ | 0.782 |
| Null model | 0.333 | $9.680\times10^{-75}$ | $1.936\times10^{-74}$ | $5.514\times10^{-75}$ | 0.000 |

**P(M)**: Denotes the prior model probabilities.

**P(M|data)**: Denotes the posterior model probabilities.

$BF_M$: Denotes the the change from prior to posterior model odds.

$BF_{01}$: Denotes Bayes factor of the best model.

$R^2$: Denotes the explained variance of each model.

**Table 10. Posterior summaries of coefficients.**

| Measures | Mean | SD | P(incl) | P(incl\|data) | BFinclusion | 95% Credible Interval | |
|---|---|---|---|---|---|---|---|
| | | | | | | Lower | Upper |
| **Religiosity** | 27.108 | 0.278 | 1.000 | 1.000 | 1.000 | 22.752 | 25.764 |
| **Mental Immunity** | 0.014 | 0.051 | 0.500 | 0.122 | 0.139 | 22.740 | 25.776 |
| **Psychological Well-being** | 1.032 | 0.058 | 0.500 | 1.000 | $2.681 \times 10^{+15}$ | 25.121 | 28.567 |

**P(incl)**: Denotes the prior inclusion probability.

**P(incl|data)**: Denotes the posterior inclusion probability.

**BF$_{inclusion}$**: Denotes inclusion Bayes factor or the change from prior to posterior inclusion odds.

diabetes due to gender in favor of males and also did not agree with the results of studies [47,48] which did not find statistically significant differences in PWB due to the gender variable.

The possible reasons for PWB less in males than females may be the nature of males in Arab societies together with coping with many stresses, as well as his diabetes and the accompanying physical symptoms that may make him fall short in performing roles expected of him, which increase stresses and negatively affects his mental health. Furthermore, the nature of males in Arab culture confirmed that the expression of sadness and pain is a weakness, all of these reasons may lead to a decrease in their PWB and MI [40,41].

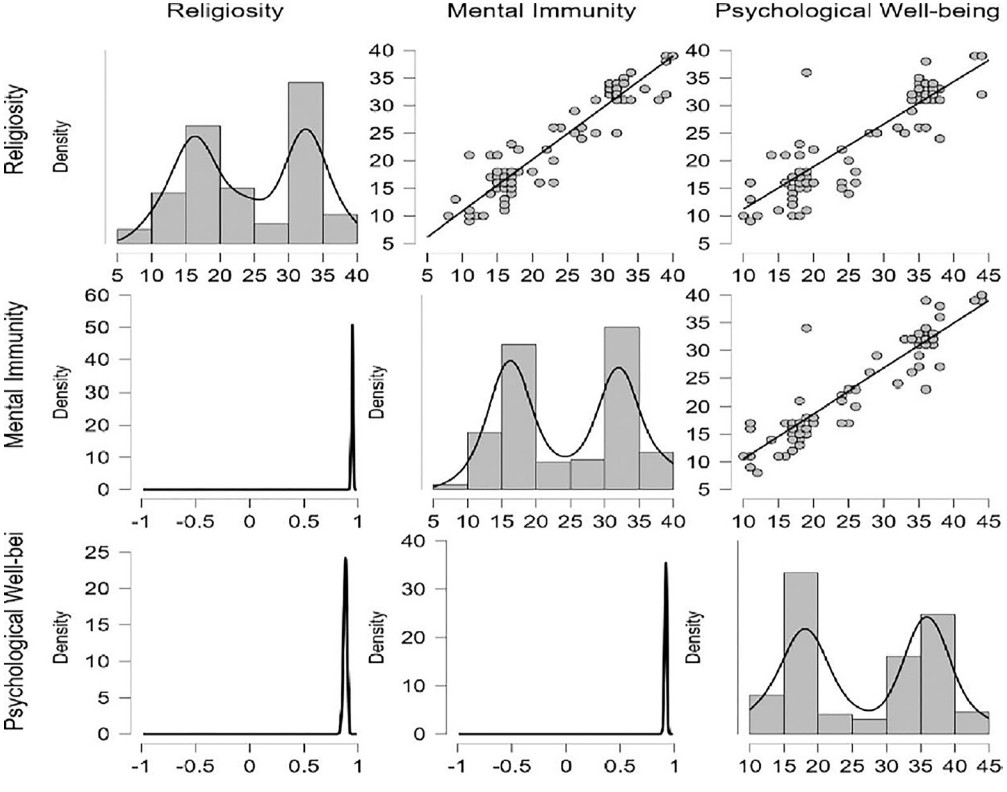

**Fig 15. Bayesian correlation matrix plot by JASP.**

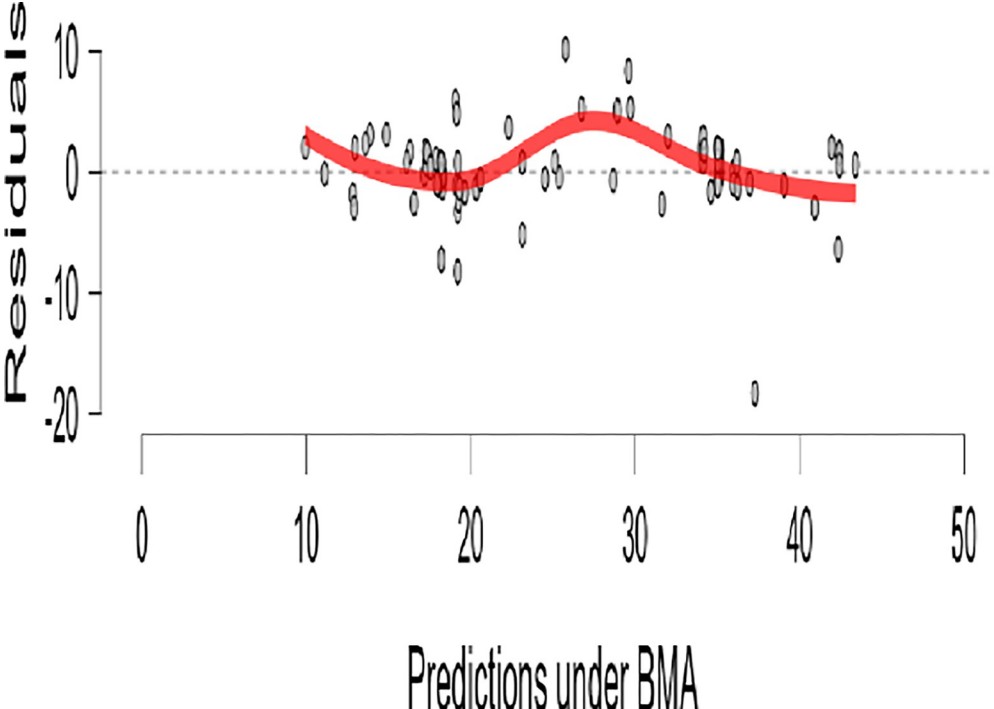

**Fig 16. Residuals vs predictions for the psychological well-being of diabetic patients by JASP.**

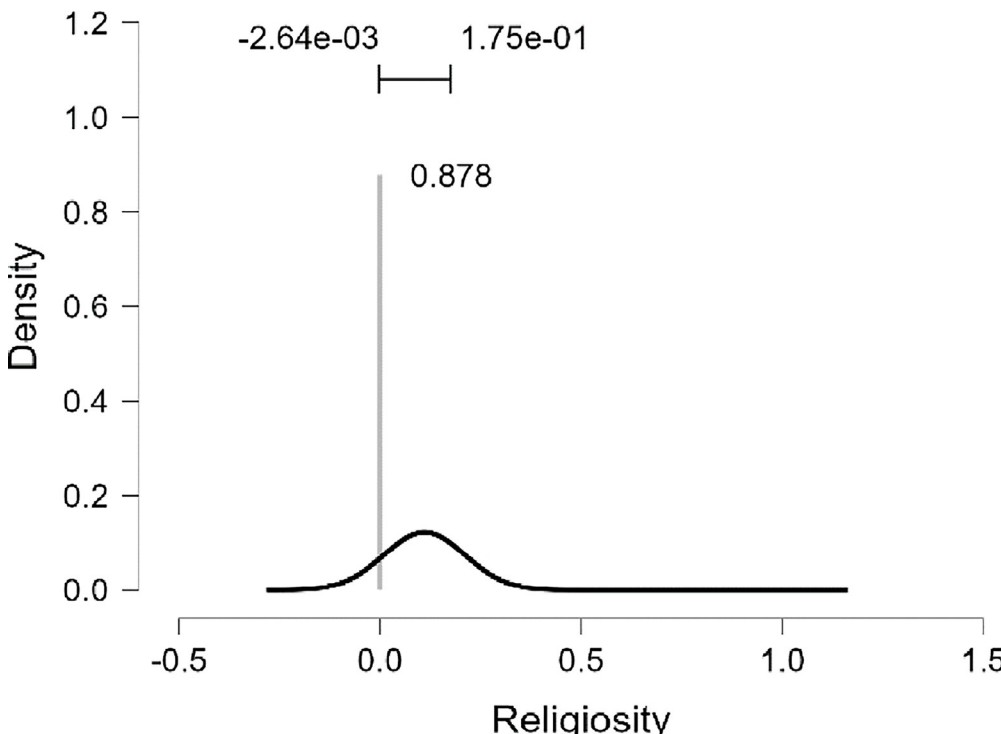

**Fig 17. Marginal posterior distributions of religiosity by JASP.**

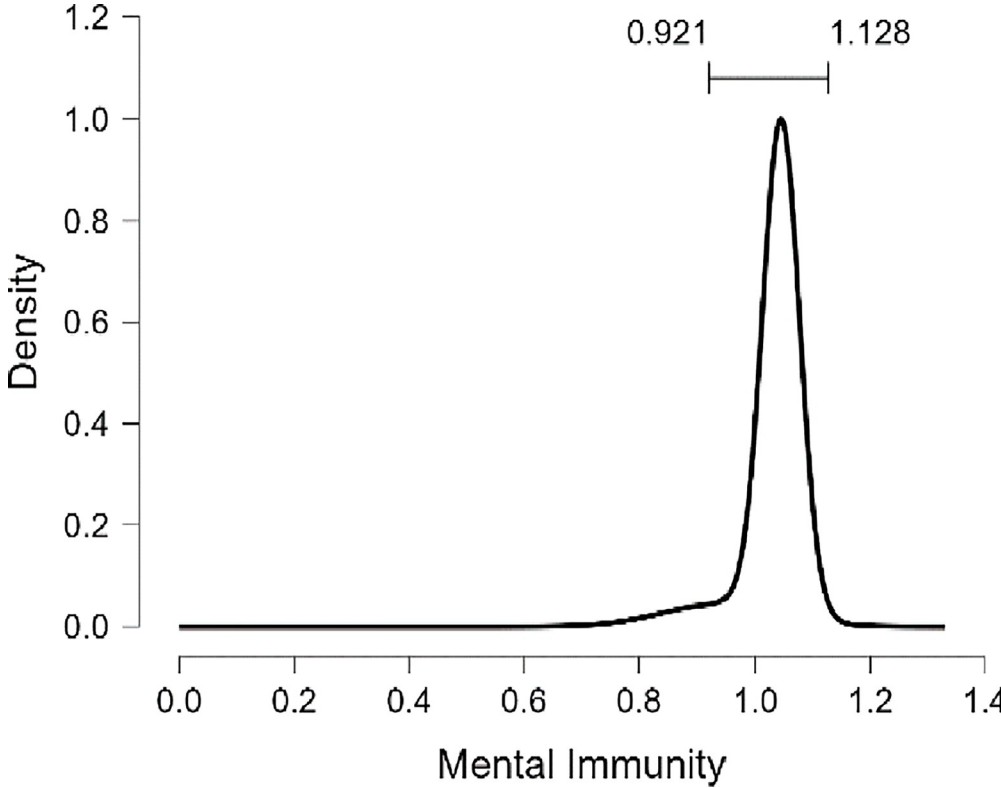

**Fig 18. Marginal posterior distributions of mental immunity by JASP.**

The males suffer more than females from chronic illness, as a result of many stresses such as raising children and taking care of the home, managing the affairs of their private life, and bearing the burdens of the disease and what it requires in terms of therapeutic management and diet, which leads to a decrease in the level of her psychological well-being, and lack of Ability to adapt to illness [42].

On the contrary, the results also revealed evidence for no differences in R, MI, and PWB among diabetic patients due to age group, with Bayesian factor values (0.176, 0.181, and 0.187) less than 1.00 and small Cohen's δ of (-0.034, -0.050, -0.063). These results are consistent with the fact that diabetes is a chronic disease with an increasing prevalence rate all over age groups around the world, leading to negative health and mental outcomes. Hence, diabetes is a health problem for humans in different age groups that harms the PWB of all patients of different ages, due to the psychological and social stresses resulting from it, as well as, the negative emotional responses [26,49,50]. For this reason, Diabetic patients are more susceptible to mental problems. Lin et al. [3] found a statistically significant correlation between negative psychological outcomes and diabetes. Moreover, previous studies [1,2,51] results indicated an association between depression and diabetes. These results may be due to the efforts made by the government of the Kingdom of Saudi Arabia to decrease the spread of diabetes and raise awareness of its harmful effects for individuals of different ages, because of the steady increase in the number of people with diabetes, and to the Kingdom of Saudi Arabia's witnessed development in the standard of living in recent times, and this has been accompanied by more luxury and dependence on fast food among a large segment of society, in addition to reliance on comfort, all of these has led to an increasing spread of diabetes to the point that it has become a danger to society.

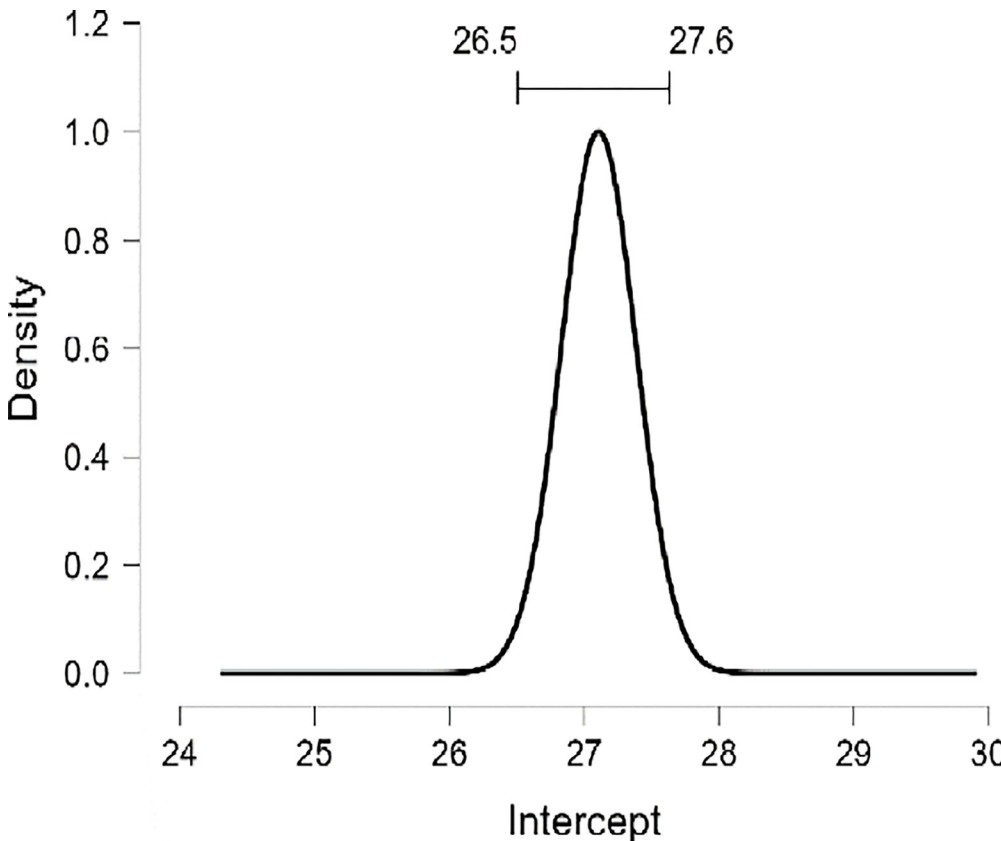

**Fig 19. Marginal posterior distributions of the interaction between mental immunity and religiosity by JASP.**

In addition, results of Bayesian linear regression analysis detected strong evidence that the model including MI is the best predictive model ($BF_{10}$ for mental immunity is 1.00 and for the other two models are 0.07 and $4.249 \times 10^{-16}$) for the PWB of diabetic patients, while there is no evidence that the model including R or the interaction between R and MI is the best predictor of PWB for diabetic patients. These findings are consistent with Person et al [52] indicating that religiosity is less predictive of healthy behavior. However, Al-Zarrouk & Masoudi [53] found a relationship between religiosity and PWB, as religiosity helps the individual accept illness and live in peace, thus leading to high levels of PWB.

Also, the results found that MI was a strong predictor of diabetic patients' PWB, and these results are consistent with the findings of the study [42] which concluded that MI predicts PWB. However these findings are not consistent with the Saffari et al. [54] study showed that people with religious and spiritual beliefs are better aligned with their diagnosis of diabetes, more compliant with drug treatment, have positive mental health indicators, and have less anxiety and depression than others. Also not consistent with Permana [55] showed that religious exercise can influence the management of diseases, including diabetes. While Sohil et al. [56] stated that religiosity is an essential element in human existence all over the world, in our study we found R is not a significant predictor of PWB in diabetes patients, but mental immunity is the best predictor.

Theoretical background about the mental immune system confirmed that R provides the individual with the ability to cope with stress and threats. In this way, MI helps diabetic

patients to develop healthy behavior and increase their ability to change conditions, while the reinforcement mechanisms involved in MI help individuals to enhance PWB [57–63].

Considering diabetes represents an increasing risk and stress on the population of the Kingdom of Saudi Arabia, as it increases the risk of amputation, nerve diseases, kidney diseases, blindness, high blood pressure, stroke, heart diseases, and dental diseases, negatively impacting the quality of life and well-being of diabetes patients [64–66], thus, we must important with screening each of R, MI, and PWB in diabetic patients. Also, according to the current study findings, if diabetic patients have high R and strong MI, they can cope effectively with stress resulting from diabetes and reduce the risks they experience, therefore, it is necessary to enhance their MI to improve their PWB.

Suffering from chronic diseases often leads to psychological trauma and a feeling of psychological stress resulting from a lifestyle change, which prompts the individual to have a disturbed behavioral response such as aggression, depression, and difficulty interacting with others, which results in introversion and a low level of self-esteem, which reduces the level of PWB among patients [52,63].

The results of the current study indicate the functionality of Bayesian statistics in verifying the differences in the study variables (R, MI, PWB) in light of gender and age variables, and the possibility of predicting the PWB of diabetics from R and MI. Arnout [67,68] mentioned the importance of Bayesian statistics in testing hypotheses in social sciences, by monitoring evidence related to the null or alternative hypothesis while accumulating data, that leads to accurate findings instead of relying on the confidence rate as in traditional frequent statistics.

## Conclusion

This study's findings indicated strong evidence supporting the differences between male and female diabetic patients in R, MI, and PWB. It found that females with diabetes have higher means of R, MI, and PWB compared to males. However, the results detected evidence supporting no differences in R, MI, and PWB among diabetic patients due to age group. Bayesian linear regression analysis revealed strong evidence that the model including MI is the best predictive model for the PWB of diabetic patients, while there is no evidence that the model including R or the interaction between R and MI is the best predictor of PWB for diabetic patients. These findings These results confirm the importance of following up on mental immunity levels among diabetic patients, and highlight the need for direct psychological care services for male diabetic patients and the urgent need to enhance IM in diabetic patients to improve their PWB. Furthermore, it is recommended for healthcare providers in Saudi Arabia to integrate MI interventions into diabetes care programs.

## Limitations and future directions

The cross-sectional design was used in the current study, which was not a follow-up, to examine the effect of diabetes on the level of R, MI, and PWB over time and at different periods. In this study, we did not examine the effect of demographic variables in predicting PWB in patients with diabetes. Also, the current study was not an interventional study that examined the effect of psychological interventions based on religiosity and mental immunity to increase PWB among diabetes patients. Therefore, we recommend the necessity of conducting longitudinal studies to investigate the impact of diabetes on PWB, R, and MI. We also recommend a future study on the role of demographic variables in the predictive model of the PWB of diabetic patients using Bayesian inference. In addition to a need to conduct a quasi-experimental intervention study based on strategies of R and MI, to increase the PWB of people with diabetes.

## Acknowledgments

Princess Nourah bint Abdulrahman University Researchers Supporting Project number (PNURSP2023R380), Princess Nourah bint Abdulrahman University, Riyadh, Saudi Arabia.

## Author Contributions

**Conceptualization:** Boshra A. Arnout, Thabit A. Al-Qahtani, Slavica Pavlovic, Mohammed R. AlZahrani, Youssef S. Abdelmotelab.

**Data curation:** Boshra A. Arnout, Slavica Pavlovic.

**Formal analysis:** Boshra A. Arnout, Abdalla S. Abdelmotelab, Youssef S. Abdelmotelab.

**Funding acquisition:** Nawal A. Al Eid.

**Investigation:** Boshra A. Arnout, Mohammed R. AlZahrani, Youssef S. Abdelmotelab.

**Methodology:** Boshra A. Arnout, Thabit A. Al-Qahtani, Abdalla S. Abdelmotelab.

**Project administration:** Nawal A. Al Eid, Boshra A. Arnout.

**Resources:** Boshra A. Arnout, Youssef S. Abdelmotelab.

**Software:** Boshra A. Arnout, Abdalla S. Abdelmotelab.

**Supervision:** Boshra A. Arnout.

**Validation:** Boshra A. Arnout.

**Visualization:** Boshra A. Arnout.

**Writing – original draft:** Boshra A. Arnout.

**Writing – review & editing:** Boshra A. Arnout, Thabit A. Al-Qahtani, Slavica Pavlovic, Mohammed R. AlZahrani, Abdalla S. Abdelmotelab, Youssef S. Abdelmotelab.

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
