## [Decision Letter · Decision Letter 0]

17 Jun 2024

PONE-D-24-08442The potential role of religiosity and psychological immunity in predicting psychological well-being of diabetic patients in Saudi Arabia within the Bayesian frameworkPLOS ONE

Dear Dr. Arnout,

Thank you for submitting your manuscript to PLOS ONE. After careful consideration, we feel that it has merit but does not fully meet PLOS ONE’s publication criteria as it currently stands. Therefore, we invite you to submit a revised version of the manuscript that addresses the points raised during the review process.

**ACADEMIC EDITOR: Kindly address following observations:**

**1. Introduction section-5th para: second line needs to be reviewed.**

**2. Page no. 3, 5th para: typing mistake (referred instead of refreed)**

**3. Hypotheses need to be revised as per APA style.**

**4. Please justify why Baysian tests were used?**

We look forward to receiving your revised manuscript.

Kind regards,

Shazia Khalid, PhD

Academic Editor

PLOS ONE

Journal Requirements:

2. e note that the grant information you provided in the ‘Funding Information’ and ‘Financial Disclosure’ sections do not match. 

[Princess Nourah bint Abdulrahman University Researchers Supporting Project number (PNURSP2023R380), Princess Nourah bint Abdulrahman University, Riyadh, Saudi Arabia. ]

 [The author(s) received no specific funding for this work.]

Reviewers' comments:

Reviewer's Responses to Questions

**Comments to the Author**

1. Is the manuscript technically sound, and do the data support the conclusions?

Reviewer #1: Partly

Reviewer #2: Yes

2. Has the statistical analysis been performed appropriately and rigorously? 

Reviewer #1: Yes

Reviewer #2: Yes

3. Have the authors made all data underlying the findings in their manuscript fully available?

Reviewer #1: Yes

Reviewer #2: Yes

4. Is the manuscript presented in an intelligible fashion and written in standard English?

Reviewer #1: No

Reviewer #2: Yes

5. Review Comments to the Author

Reviewer #1: Formatting Issues:

Ensure proper punctuation after headings, such as a colon.

Capitalize the first letter of scale names in the method section.

Ensure that Table 4 appears on a single page.

Literature Review:

Enhance the literature review by incorporating additional research to provide a more comprehensive analysis.

Strengthen the argumentative and analytical aspects of the literature review to improve coherence and logical flow.

Introductory Paragraph:

Revise the introductory paragraph to provide a clearer overview of the research topic, objectives, and methodology.

Bayesian Model Integration:

Clarify the integration of the Bayesian model into the research methodology, particularly in relation to measuring religiosity and immunity in the Kingdom of Saudi Arabia, despite including non-Saudi populations in the sample.

Consistency in Draft:

Correct the consistent error in writing t-test names by ensuring the first letter is lowercase and italicized throughout the draft.

Check for typographical errors, such as missing letters, in figures and throughout the text.

In-text Citation Consistency:

Standardize the citation format throughout the document to ensure consistency. Decide on a specific format, such as APA or MLA, and apply it uniformly.

Discussion Section:

Expand and deepen the discussion section to provide a more thorough analysis of the research findings and their implications.

In summary, the document requires revisions to address formatting issues, enhance the literature review, clarify the integration of the Bayesian model, maintain consistency throughout the draft, and strengthen the discussion section.

Reviewer #2: Overall the research paper impressively investigates into a topic but I would like to share few recommendations.

1. Paper has included many theories relevant to the study variables but I would suggest to summarize these theories and also specify which one theory you are focusing on while working on your research.

2. The paper's quality is elevated by the careful attention given to its methodology; however, the details of sampling technique used in the research will be a valuable addition. Also the details of how the assessment measure were developed by the researchers of the study should also be included.

3. Rationale and objectives of the study are missing. It must be included and the reason for using Bayesian framework should also be highlighted specifying its advantages and disadvantages.

4. The tile looks fine but it is just a recommendation after reading the whole paper that it should be rephrased and the role of gender and age should be incorporated in the title which are related to the main objectives of the study.

5. Future recommendations should also be added in the paper.

6. PLOS authors have the option to publish the peer review history of their article (what does this mean?). If published, this will include your full peer review and any attached files.

Reviewer #1: No

Reviewer #2: No

---

## [Author Response · Author response to Decision Letter 0]

21 Jun 2024

Dear Editor

Greetings

Thank you for your great efforts and the reviewers, whose comments contributed to enhancing our manuscript. All comments have been taken into account, and We have edited all of the reviewer comments, we hope that the revised manuscript meets your acceptance.

Best regards

1. Formatting Issues:

Ensure proper punctuation after headings, such as a colon.

Capitalize the first letter of scale names in the method section.

Ensure that Table 4 appears on a single page.

Reply: 

Thank you for your important comment, we edited all of these and highlighted them with yellow.

2. Literature Review:

Enhance the literature review by incorporating additional research to provide a more comprehensive analysis. Strengthen the argumentative and analytical aspects of the literature review to improve coherence and logical flow.

Reply: 

Thank you for your important comment, we edited and highlighted it with yellow.

Introductory Paragraph:

Revise the introductory paragraph to provide a clearer overview of the research topic, objectives, and methodology.

Reply: 

Thank you for your important comment, we edited and highlighted it with yellow.

Bayesian Model Integration:

Clarify the integration of the Bayesian model into the research methodology, particularly in relation to measuring religiosity and immunity in the Kingdom of Saudi Arabia, despite including non-Saudi populations in the sample.

Reply: 

Thank you for your important comment, we edited this in the data analysis section and highlighted it with yellow.

Consistency in Draft:

Correct the consistent error in writing t-test names by ensuring the first letter is lowercase and italicized throughout the draft.

Reply: 

Thank you for your important comment, we corrected it in the manuscripts and highlighted all of them in yellow.

Check for typographical errors, such as missing letters, in figures and throughout the text.

In-text Citation Consistency:

Reply: 

Thank you for your important comment, we corrected all and highlighted all of them in yellow.

Standardize the citation format throughout the document to ensure consistency. Decide on a specific format, such as APA or MLA, and apply it uniformly.

Reply: 

Thank you for your important comment, we checked it all and highlighted it with yellow.

Discussion Section:

Expand and deepen the discussion section to provide a more thorough analysis of the research findings and their implications.

Reply: 

Thank you for your important comment, we edited and highlighted it with yellow.

In summary, the document requires revisions to address formatting issues, enhance the literature review, clarify the integration of the Bayesian model, maintain consistency throughout the draft, and strengthen the discussion section.

Reply: 

We are grateful for your valuable feedback that has contributed to improving the quality of our manuscript. All comments have been taken into account, and we edited the comments and hope they meet with your approval.

Reply to the comments of the reviewer 2: 

Overall the research paper impressively investigates into a topic but I would like to share few recommendations.

1. Paper has included many theories relevant to the study variables but I would suggest to summarize these theories and also specify which one theory you are focusing on while working on your research.

Reply: 

Thank you for your important comment, we summarized it and wrote that we developed the PWB scale according to the PWB Ryff model, and highlighted it with yellow.

2. The paper's quality is elevated by the careful attention given to its methodology; however, the details of sampling technique used in the research will be a valuable addition. Also the details of how the assessment measure were developed by the researchers of the study should also be included.

Reply: 

Thank you for your important comment, we edited and highlighted it with yellow.

3. Rationale and objectives of the study are missing. It must be included and the reason for using Bayesian framework should also be highlighted specifying its advantages and disadvantages.

Reply: 

Thank you for your important comment, we added reasons for using the Bayesian framework and highlighted it with yellow.

4. The tile looks fine but it is just a recommendation after reading the whole paper that it should be rephrased and the role of gender and age should be incorporated in the title which are related to the main objectives of the study.

Reply: 

Thank you for your important comment, we edited the study title according to this recommendation and highlighted it with yellow.

5. Future recommendations should also be added in the paper.

Reply: 

Thank you for your important comment, we wrote it in the (limitations and future directions) section and highlighted it with yellow.

Dear reviewer, 

We are grateful for your valuable feedback that has contributed to improving the quality of our manuscript. All comments have been taken into account, and we edited the comments and hope they meet with your approval.

Journal Requirements:

Reply: 

Thank you for your important comment, we edited the manuscript according to PLOS ONE's style requirements. and highlighted it with yellow.

2. e note that the grant information you provided in the ‘Funding Information’ and ‘Financial Disclosure’ sections do not match. 

Reply: 

Thank you for your important comment, we edited and highlighted it with yellow.

This work was supported by Princess Nourah bint Abdulrahman University Project number (PNURSP2023R380). 

[Princess Nourah bint Abdulrahman University Researchers Supporting Project number (PNURSP2023R380), Princess Nourah bint Abdulrahman University, Riyadh, Saudi Arabia. ]

 [The author(s) received no specific funding for this work.]

Reply: 

Thank you for your important comment, please, our university demanded the following format of the Acknowledgment Section, necessary remain as it is : 

Princess Nourah bint Abdulrahman University Researchers Supporting Project number (PNURSP2023R380), Princess Nourah bint Abdulrahman University, Riyadh, Saudi Arabia. 

Reply: 

Thank you for your important comment, we reviewed the references and edited and highlighted them with yellow.

---

## [Editor Report · Decision Letter 1]

9 Jul 2024

PONE-D-24-08442R1The potential role of religiosity, psychological immunity, gender, and age group in predicting the psychological well-being of diabetic patients in Saudi Arabia within the Bayesian frameworkPLOS ONE

Dear Dr. Arnout,

The manusrcipt is approved for publication after incorporating following changes:1. Delete Null Hypotheses.2. Include reference for line no. 149.3. Reword line no. 219 as 'factorial structure' instead of 'structural factorial.'

Please submit your revised manuscript by Aug 23 2024 11:59PM. If you will need more time than this to complete your revisions, please reply to this message or contact the journal office at plosone@plos.org. Please include the following items when submitting your revised manuscript:A rebuttal letter that responds to each point raised by the academic editor and reviewer(s). You should upload this letter as a separate file labeled 'Response to Reviewers'.A marked-up copy of your manuscript that highlights changes made to the original version. You should upload this as a separate file labeled 'Revised Manuscript with Track Changes'.An unmarked version of your revised paper without tracked changes. You should upload this as a separate file labeled 'Manuscript'.If applicable, we recommend that you deposit your laboratory protocols in protocols.io to enhance the reproducibility of your results. Protocols.io assigns your protocol its own identifier (DOI) so that it can be cited independently in the future. For instructions see: https://journals.plos.org/plosone/s/submission-guidelines#loc-laboratory-protocols. Additionally, PLOS ONE offers an option for publishing peer-reviewed Lab Protocol articles, which describe protocols hosted on protocols.io. Read more information on sharing protocols at https://plos.org/protocols?utm_medium=editorial-email&utm_source=authorletters&utm_campaign=protocols.

We look forward to receiving your revised manuscript.

Kind regards,

Shazia Khalid, PhD

Academic Editor

PLOS ONE
---

## [Author Response · Author response to Decision Letter 1]

10 Jul 2024

Dear Editor 

Greeting 

Thank you for your revision of our manuscript. We edited all changes and highlighted it in yellow. We hope the revised manuscript receive your acceptance. 

Best regards

 The manusrcipt is approved for publication after incorporating following changes:

1. Delete Null Hypotheses.

Thank you for this comment, we edited it. 

2. Include reference for line no. 149.

Thank you for this comment, we edited it. 

3. Reword line no. 219 as 'factorial structure' instead of 'structural factorial.'

Thank you for this important comment, we edited it. 

Best regards

---

## [Editor Report · Decision Letter 2]

24 Jul 2024

The potential role of religiosity, psychological immunity, gender, and age group in predicting the psychological well-being of diabetic patients in Saudi Arabia within the Bayesian framework

PONE-D-24-08442R2

Dear Author,

We’re pleased to inform you that your manuscript has been judged scientifically suitable for publication and will be formally accepted for publication once it meets all outstanding technical requirements.

Kind regards,

Shazia Khalid, PhD

Academic Editor

PLOS ONE

---

## [Editor Report · Acceptance letter]

14 Aug 2024

PONE-D-24-08442R2 

PLOS ONE

Dear Dr. Arnout, 

I'm pleased to inform you that your manuscript has been deemed suitable for publication in PLOS ONE. Congratulations! Your manuscript is now being handed over to our production team.

Kind regards, 

on behalf of

Professor Shazia Khalid 

Academic Editor

PLOS ONE